# Nature-Based Meditation, Rumination and Mental Wellbeing

**DOI:** 10.3390/ijerph19159118

**Published:** 2022-07-26

**Authors:** Matthew Owens, Hannah L. I. Bunce

**Affiliations:** 1Department of Psychology, The Mood Disorders Centre, University of Exeter, Exeter EX4 4QQ, UK; 2The ROWAN Group, Exeter EX4 4QQ, UK; hb616@exeter.ac.uk; 3CEDAR, University of Exeter, Exeter EX4 4QQ, UK; 4Somerset Foundation Trust NHS, Taunton TA1 5DA, UK

**Keywords:** nature, depression, wellbeing, rumination, nature-based intervention, meditation

## Abstract

Novel approaches for children and young people (CYP) in the prevention and intervention of mental illness are needed and nature-based interventions (NBI) may be clinically useful. This proof-of-principle study tested the effects of a novel brief nature-based meditation on rumination, depressive symptoms and wellbeing in young people. Sixty-eight university students were randomised to one of three conditions: active control (n = 23), indoor meditation (n = 22) or nature-based meditation (n = 23). Participants completed self-report measures on state and trait rumination post intervention and depression and wellbeing at a 2-week follow-up. Depressive rumination significantly decreased post intervention in the nature condition and depressive symptoms improved for both intervention groups. Wellbeing only significantly improved at follow-up in the nature condition. Nature condition participants demonstrated one minimal clinically important difference (MCID) for wellbeing at follow-up. Depressive symptoms for this condition were below the clinically significant threshold for depression. The number needed to treat (NNT) analysis suggested that two to five young people would need to complete the intervention. Preliminary evidence suggests NBIs, such as the one in the present study, can reduce depressive rumination and symptoms and improve wellbeing. Replication with larger clinical samples is required to substantiate findings.

## 1. Introduction

Common mental health problems such as clinical depression are highly prevalent in Europe (6.9%) [1] and worldwide (4.4.%) [2]. The situation has been exacerbated recently for young people following the response to the COVID-19 pandemic [3,4], where the global prevalence of depression has been estimated at ~30% [5,6]. Mental health problems are complex and have their roots in childhood and adolescence. This is illustrated by the fact that approximately 50% of mental health disorders emerge in the teenage years [7]; 75% by 24 years of age [8]. In addition, common mental health difficulties such as depression and anxiety may have been rising over the past decade [9], despite the efforts of evidence-based treatments [10]. The psychosocial sequelae of mental health difficulties and low wellbeing are many [11] and include recurrence of mental ill health [12], increased risk of failure to complete secondary education [13], labour market marginalisation [14] and somatic illness [15]. Furthermore, there are huge economic costs attached to mental health difficulties [16].

As a society, we tend to suffer from a health care myopia, often tending to focus on the “rule of rescue” rather than more long-term prevention approaches, which are more cost-effective and arguably more ethical to deliver [17]. While treatment can be effective for many, there is a growing recognition that there are limitations to a treatment-only model [18], including perceived barriers to services [19]. For example, stigma, lack of accessibility (time, transport, cost), fear or stress about the act of help-seeking or source of help itself, not wanting to be a burden on others and a preference for self-reliance are all obstacles in gaining mental health support [20]. Unfortunately, this means that most young people in need of treatment for mental health problems do not actually receive any [21], and even assuming a hypothetical scenario with no barriers to treatment and complete coverage and compliance, it has been estimated that only around 36% of the burden of depression could be averted, depending on knowledge and interventions [22]. In cases where treatment is sought, low responses and high relapse rates highlight the limitations of treatment-only approaches to mental health [23]. Considering these challenges and the pressure that mental health services have come under in recent years, the focus on prevention interventions has become increasingly urgent [24]. The scale of the problem is illustrated by a five-fold increase in university students reporting mental health conditions from 2007 to 2018 [25]. In the U.K., a number of policies have placed particular emphasis on prevention interventions as a priority in improving mental health provision for CYP, including the Five Year Forward View [26] and Future in Mind [27].

Meta-analytic evidence suggests that depression can be prevented using traditional psychological therapeutic approaches. For example, van Zoonen et al. [28] showed that the incidence of depression was reduced by approximately 21% for those in an intervention group (primarily cognitive behavioural therapy: CBT, or interpersonal therapy: IPT), with around 20 people requiring treatment in order to prevent a single depression case. Recent evidence suggests that the prevention of common mental health problems is achievable in young people [29], including college students [30]. Furthermore, evidence shows that self-help approaches are also efficacious in alleviating symptoms in CYP. For instance, a review and meta-analysis of self-help interventions showed that guided and unguided approaches (e.g., online CBT, bibliotherapy) were beneficial when compared to control, and only slightly less effective than face-to-face treatment [31]. Other promising prevention approaches include those targeting lifestyle factors such as diet, sleep and exercise [32]. More recently, novel approaches such as increasing contact and connection with natural environments have been tested in the research literature [33].

Natural environments are thought to confer a myriad of benefits to human health [34,35,36], both mental [37] and physical [38]. For example, a number of nature-based interventions have demonstrated reliable reductions in stress [39]. Similarly, the Japanese art of shinrin yoku, or forest bathing (which includes an inherent mindfulness component), has produced reliable reductions in self-reported stress, cortisol [40,41] and depressive symptoms [42]. Emerging evidence therefore suggests that nature-based interventions (NBI) may be one way to bring benefit to those with common mental health difficulties such as depression [43,44,45]. However, the core mechanisms that underly the putative beneficial effect of NBIs on mental wellbeing are currently poorly understood. Several candidates have been posited in the literature [46], two of which, mindfulness and rumination, are the focus of the present study.

Depressive rumination, defined as the tendency to repetitively analyse the causes, meanings and consequences of problems and symptoms of depression [47], is a well-established and important mechanism in depression which has been robustly implicated in its onset and maintenance [48,49]. Common foci of rumination include social interactions and personal relationships, past mistakes and negative experiences [50]. A novel approach to CBT for depression has been developed that targets rumination (rumination-focussed cognitive-behavioural therapy; RFCBT), which has been shown to be effective as both a treatment [51] and a prevention of depression [52], as well as a self-help intervention [53].

Mindfulness has been defined as “paying attention in a particular way: on purpose, in the present moment, and non-judgementally” [54]. Mindfulness-based interventions (MBI) are efficacious for treating depression [55]. MBIs have demonstrated non-inferiority to both pharmacological treatment in preventing a relapse or the recurrence of depression [56] and psychological evidence-based treatments in reducing a range of psychiatric disorder symptoms [57]. MBI for depression are recommended by the National Institute for Clinical Excellence [58]. Mindfulness-Based Cognitive Therapy (MBCT) has recently been shown to be effective in reducing depressive symptoms when delivered in a real-world setting (primary care) [59]. Interestingly the clinical mechanisms of action in MBCT and Mindfulness-Based Stress Reduction (MBSR) on mental health and wellbeing include cognitive and emotional reactivity, rumination and worry [60]. Furthermore, brief MBIs, including home practice, have been shown to reduce symptoms of depression and rumination in depressed patients, relative to control [61].

Mindfulness interventions in the context of natural environments have been shown to be effective in improving health and wellbeing. For example, a systematic review including a range of mindfulness NBIs from short single-session interventions to residential forest bathing programmes to a 16-week programme including MBCT found positive changes on psychological, physical and social outcomes [62]. In a randomised experiment, Nisbet and colleagues found that participants in a 20 min outdoors walking condition that included instructions to engage in mindfulness reported less negative affects than those randomised to either walking indoors or walking outdoors without meditation [63]. Several randomised controlled trials (RCT) have tested mindfulness approaches against standard treatment. For example, Stigsdotter et al. [64] compared a 10-week NBI treatment for depression developed using principles of mindfulness-based stress reduction (MBSR) and CBT, also drawing on principles from environmental psychology with a standard CBT intervention. The authors found that both interventions improved psychological wellbeing over a 12-month follow-up period.

Further trials incorporating mindfulness in interventions have also tested whether depressive rumination is a modifiable target. In a pragmatic controlled trial, McEwan et al. [65] compared the effect of compassionate mind training, forest bathing and a group combining the two interventions on a range of outcomes including depressive rumination. The study results showed improvements for all groups on self-report measures at 3-month follow-up, including reduced depressive symptoms, reduced rumination on problems and increased positive emotions. In a randomised controlled trial Choe et al. [66] tested whether brief MBSR (6 weeks) as a wellbeing intervention carried out in different settings (indoor, built outdoor and natural outdoor) would be enhanced when combined with exposure to natural environments at 1-week and 3-month follow up. The results suggested that improvements in wellbeing were made for both the built outdoor and natural outdoor groups, relative to indoor on some measures (e.g., positive affect) and more so, particularly, for the natural outdoor group on others (e.g., rumination and stress).

Furthermore, the results of several studies suggest that reductions in rumination may indeed be one way in which exposure to natural environments can lead to better mental wellbeing. For example, after inducing rumination in adults diagnosed with clinical depression, positive affects increased for participants randomised to a nature walk but not for those randomised to an urban walk [67]. Similarly, Bratman et al. [68] showed that walking in nature can reduce rumination relative to urban walking and this effect may be partly explained by reduced activity in the subgenual prefrontal cortex, an area implicated in depressive rumination [68,69]. Recently, Bratman et al. [70] provided preliminary evidence in cross-sectional data to suggest that the association between spending time in nature and negative effects could be mediated by reductions in the tendency to ruminate.

### The Present Study

The aim of this proof-of-principle study was to test the ability of a novel brief 20-min nature-based self-guided meditation to reduce rumination in the short-term, increase mental wellbeing and reduce depressive symptoms over a two-week period in a vulnerable population (university students). To do this, we compared our nature-based meditation intervention delivered in a natural environment with both an active control and standard non-nature-based indoor meditation.

## 2. Materials and Methods

The participants were 68 university undergraduate or postgraduate students, 18 years or older (mean = 20.93, SD = 3.61; 26 males (38.24%), 42 females (61.75%); 47 international students (69.12%) and 21 domestic (30.88%).

### 2.1. Sample Size

We based our a priori sample size calculation on the study by Choe et al. [65] on the psychological effects of a 6-week MBSR programme in three settings (indoor, outdoor urban and natural outdoor). In this study the authors report a significant interaction effect between time and environmental setting, where rumination was lowered in all groups, particularly in the natural outdoor group. The effect size (partial eta squared) for the Group × Time interaction in the ANOVA reported was η^2^ = 0.09. We reasoned that an effect on reducing rumination of this magnitude would be worth detecting. We calculated that to have 80% power to detect a similar effect size (i.e., η^2^ = 0.09), a sample size of 105 would be required. In the event, we recruited 68 participants to the study, which had the effect of reducing the power in the study to 59%. The retention rate at two-week follow up was 88%.

### 2.2. Hypotheses

The study was designed to test the following hypotheses: (i) A brief audio-recorded self-guided meditation will reduce rumination over the short term relative to an active control. (ii) Self-reported depressive symptoms will be reduced and wellbeing increased after a two-week period of self-guided meditation practice, again relative to the control group. (iii) We expected the nature-based meditation to be superior to the non-nature-based meditation on all outcomes. (iv) We predicted a dose–response relationship between the number of reported meditation attempts and any change detected in depressive symptoms or wellbeing after the two-week period.

This study was reviewed and approved by the Psychology Ethics Committee at the University of Exeter (reference number: 494103).

### 2.3. The Nature-Based Meditation Condition

We designed a brief 20 min nature-based meditation for this study, created as an audio file to be listened to independently via a portable device with headphones. The meditation is underpinned by key theories and approaches used in clinical psychology; mindfulness, Compassion-Focussed Therapy, mental imagery and forest bathing. The meditation begins and ends with a brief grounding breathing meditation, drawn from mindfulness-based approaches to firstly focus the individual into the present moment and to facilitate the practice of intentionally paying attention, non-judgementally [54], particularly to nature surroundings. After a period of mindful breathing, listeners were guided to purposefully slow their breathing rate down and adopt a soothing rhythm, a technique used in Compassion Focused Therapy [71]. Evidence suggests that breathing in this way stimulates the parasympathetic nervous system, thereby encouraging a state of relaxation [72]. In alignment with forest bathing principles, the meditation supported listeners to explore and value their natural surroundings through guided exercises [65]. Listeners were invited to engage their senses as they slowly walked around (for example “begin to notice the sounds of nature around you”). The meditation asked listeners to discover and play with concepts such as light and shadow and observe nature elements both near and far. Using mental imagery techniques with the aim of facilitating nature connection and amplifying the potential positive emotional effects of the meditation, the listener was prompted to imagine nature images, (for example, “feeling rooted to the ground”). Imagery facilitates seeing as if “seeing with the mind’s eye” [73] and elicits stronger emotional responses than their verbal counterparts and can act as an “emotional amplifier” [74].

### 2.4. The Indoor Meditation Condition

The indoor meditation matched the nature-based meditation in duration and modality (aural task). Nature words and references were replaced with indoor substitutes, (for example, “Can you feel the texture of the floor under your feet?”). The beginning and end of the meditation mirrored the nature-based meditation, using mindfulness breathing techniques. Soothing rhythm breathing followed this before leading into guided exercises, such as for the nature condition, including using the senses to explore the indoor space.

### 2.5. The Active Control Condition

The active control was designed to match the other condition tasks in duration and in modality. Participants were asked to listen to an audio tour guide whilst walking around the University “Forum”. The Forum is a busy, large communal building consisting of retail shops, coffee and food shops, a bar, study spaces and a library. There is also a central meeting mezzanine space. Participants were guided at a slow pace around the Forum, listening independently to the audio file. They were informed beforehand that they would be asked questions about the Forum after the session to encourage adherence to the task. Eight binary yes/no adherence questions were set (e.g., “Does the Music and Drama room-open from 7 a.m. to 6 p.m.?”) and adherence to the task was deemed met with ≥60% correct answers (i.e., above chance level = 5/8). The locations of the experimental conditions are illustrated in Figure 1.

### 2.6. Measures

#### 2.6.1. Ruminative Response Styles (RRS)

We used the five-item Brooding subscale of the Ruminative Response Scale [75] to measure rumination. Participants are asked to respond to five statements (e.g., “what am I doing to deserve this”) using a five-point scale (1 ‘almost never’ to 4 ‘almost always’). The scale has been shown to be reliable in previous research [76] and the internal consistency (α) in the present sample was good at T0 (α = 0.82) and T1 (α = 0.80).

#### 2.6.2. Brief Rumination State Inventory (BSRI)

The Brief State Rumination Inventory (BSRI) is a psychometrically valid measure sensitive to situational changes in rumination [77]. The scale has 8 items (e.g., “right now, it is hard for me to shut off negative thoughts about myself”. Participants indicate their response on a Visual analogue scale (VAS) ranging from 0 (Completely disagree) to 100 (Completely agree). Participants’ scores are summed to give an overall score. In the current study scale, reliability ranged from α = 0.87 to 0.91 at T0 and T1, respectively.

#### 2.6.3. Patient Health Questionnaire (PHQ-8)

We used the Patient Health Questionnaire PHQ-8 [78] in the current study, a well-validated eight-item self-report instrument that assesses current symptoms of depression over the last two weeks based on DSM-IV criteria for major depression. The PHQ-8 compares well with the original PHQ-9 [79]. The internal consistency (α) in the present study was good at T0 (0.88) and T2 (0.85). The minimal clinically important difference (MCID) has been estimated as 2–3 points [80] and we considered 3 points to indicate an MCID in the present study. A score of lower than 5 indicates only minimal depression [81].

#### 2.6.4. Short Warwickshire Edinburgh Mental Wellbeing Scale (SWEMWBS)

The Warwick Edinburgh Mental Wellbeing Scale (WEMWBS) is a well-validated measure of wellbeing, suitable for young people [82] and sensitive to change in mental wellbeing prevention interventions [83]. We used the short 7-item version in the present study [84]. The scale asks participants to reflect over the last two weeks (e.g., “I have been feeling optimistic about the future”) and responses are coded using a 5-point scale ranging from 1 “none of the time” to 5 “all of the time”, scores range from 1 to 35. The WEMWBS has demonstrated good internal reliability in student samples [85] and test–retest reliability over one week is high [84]. In the current study scale, reliability was good at T0 (α = 0.86) and T2 (α = 0.88). The MCID for the SWEMWBS is 3 points [86] and a score of 27.5 or higher indicates high wellbeing [87].

### 2.7. Statistical Analysis

Statistical analysis was carried out in SPSS (version 28) (IBM, Chicago, IL, USA) and STATA (version 17) (StataCorp, College Station, TX, USA). We tested our hypotheses relating to the effect of the meditations on mental wellbeing over time (i to iii) using repeated measures analysis of variance (RMANOVA). To test whether the number of meditation attempts was associated with the two-week outcome measures (PHQ-8 and SWEMWBS), we used a linear regression model with meditation attempts as a predictor of T2 outcomes, adjusting for baseline scores. We report partial eta squared (η^2^) following the results of the RMANOVA and use the following conventions for interpreting the effect size: 0.01 = small; 0.06 = medium; ≥0.14 = large. The Gaussian distribution of variables was inspected using Q-Q plots and formally tested using the Shapiro–Francia test (W′), shown to perform well compared to a range of normality tests [88]. Tukey’s ladder of powers was used to select the optimal transformation. The normality of the distribution in the standardised residuals was tested for each regression model in the adherence analysis. To confirm that any associations detected were not caused by undue influence of individual scores, two further post hoc diagnostic tests were performed. Cook’s D, a measure of an individual observation’s effect on overall fit and DFBETA, a measure of individual influence on a given beta. In both cases a score of <1 indicated no undue influence of individual scores. To estimate the number needed to treat (NNT) for either meditation intervention, we used the MCID for depressive symptoms and wellbeing to create categories of change for each participant. In this way the absolute risk reduction (ARR) can be calculated and used for meaningful clinical change over the two-week period. The NNT is calculated using the following equation:$$\mathrm{NNT}=\frac{1}{\mathrm{ARR}}$$

### 2.8. Procedure

Participants were recruited for the study using flyers and posters on the university campus and through email through the local University research recruitment platform and through the use of social media (local Facebook group and Instagram). Participants read the study information sheets online and completed the initial screening housed in the online survey platform, Qualtrics. Eligible participants were then randomised by the research assistants using a coded block-randomised (1:1) list (e.g., B,B,C,A,C,B, etc.), which was generated by the P.I. (MO) prior to the recruitment phase using sealedenvelope.com (accessed on 21 November 2021). The P.I. was blinded to the random allocation process and did not conduct any of the group sessions to reduce bias in the analyses. Participants were contacted by the research assistants to arrange attendance to a session in either the control, indoor or nature meditation groups. Two research assistants led each session across all three groups. There were no more than 6 participants in each group. For the control group, the participants met the research assistants outside the University Forum and were instructed to listen to the audio file using headphones and to be guided around the Forum. Before (T0) and after (T1) the single session, participants completed a battery of questionnaires in Qualtrics. After the session, the meditation groups were encouraged to practice the meditations in their own time for a minimum of three times over the course of the following 2 weeks. All participants were sent the final assessment battery 2 weeks after the group session (T2). Participants were entered into a prize draw. Winners were randomly selected for one of three small monetary prizes: 1 × £10, 1 × £20 and 1 × £30 in vouchers.

## 3. Results

The flow of participants through the study is shown in Figure 2. Of the 99 young people that initially completed the online survey, 31 declined to participate or were not contactable. A total of 68 participants were randomised to one of the three conditions.

The PHQ-8 was the only variable (at T0 and at T2) that did not follow a Gaussian distribution. Both were improved following a square root transformation (see Table 1). There were no differences between the conditions on sex (x^2^ = 4.41, df(2), *p* = 0.13), international/domestic student status (x^2^ = 1.64, df(2), *p* = 0.44), age (F(2.65) = 0.70, *p* = 0.50) or any of the baseline measures (See Table 2), suggesting that the randomisation process was successful. All participants in the control condition were deemed to have adhered to the task (Mean correct = 6.78, SD = 1.04; range = 5–8). The average duration in the study was 17.73 days (SD = 3.95; range 14–28). 

We report the PHQ-8 analysis in this section, leaving the variables in their original form, given that it is well-known and validated scale, with a readily interpretable metric. For completeness, the results of the RMANOVA Group × Time interactions using the transformed PHQ-8 variables are also reported. RRS = Ruminative Response Styles; BSRI = Brief Rumination State Inventory; PHQ-8 = Patient Health Questionnaire; Short Warwickshire Edinburgh Mental Wellbeing Scale; W′= Shapiro–Francia test.

### 3.1. Immediate Change in Rumination

We found a significant main effect of Time for RRS, which decreased over time (F(1,64) = 6.23, *p* = 0.02, η^2^ = 0.09), and a significant Group × Time interaction (F(2,64) = 3.75, *p* = 0.03, η^2^ = 0.11). Follow-up multivariate tests indicated a significant reduction in RRS in the nature condition (F(1,22) = 8.17, *p* = 0.001, η^2^ = 0.27) but not in the indoor (F(1,21) = 0.50, *p* = 0.49, η^2^ = 0.02) or control conditions (F(1,21) = 2.61, *p* = 0.12, η^2^ = 0.11). See Figure 3. There was a significant main effect of time on the BSRI which also indicated a reduction in state rumination over time (F(1,65) = 7.77, *p* < 0.01, η^2^ = 0.11) but no Group × Time interaction, suggesting no differences across conditions (F(2,65) = 3.75, *p* = 0.09, η^2^ = 0.07). See Table 1 for details.

### 3.2. Depression Symptoms and Mental Wellbeing

#### 3.2.1. Depressive Symptoms

There was a significant main effect of Time, indicating an average improvement in depressive symptoms over the two-week follow up period (F(1,57) = 12.38, *p* < 0.001, η^2^ = 0.18), which was qualified by a significant Group × Time interaction (F(2,57) = 3.34, *p* = 0.04, η^2^ = 0.11; (transformed analysis: F(2,57) = 3.66, *p* = 0.03, η^2^ = 0.11)). Follow up multivariate tests indicated that there was a significant reduction in the PHQ-8 in both the nature condition (F(1,57) = 13.94, *p* < 0.001, η^2^ = 0.20) and the indoor condition (F(1,57) = 5.50, *p* = 0.02, η^2^ = 0.09) but not the control condition (F(1,57) = 0.00, *p* = 0.96, η^2^ = 0.00); see Figure 4 for an illustration. The nature condition was alone in showing a MCID (3.67) and the average PHQ-8 score fell below 5 (minimal depression).

The NNT was 3.0 and 3.7 in the nature and indoor meditation groups, respectively, indicating that 3 to 4 young people would need to receive the intervention for one additional person to show 1 MCID in depressive symptoms.

#### 3.2.2. Wellbeing

The main significant effect was of Time, suggesting an increase in subjective mental wellbeing for the whole group over the two-week follow up period (F(1,57) = 23.42, *p* < 0.001, η^2^ = 0.29). However, a significant Group × Time interaction (see Figure 5) suggested that the improvement was not uniform across conditions (F(2,57) = 9.19, *p* < 0.001, η^2^ = 0.24). Follow up multivariate tests indicated that the significant increase in scores on the SWEMWBS was only found in the nature condition (F(1,57) = 40.01, *p* < 0.001, η^2^ = 0.41) and not in the indoor condition (F(1,57) = 2.77, *p* = 0.10, η^2^ = 0.05) nor the control condition (F(1,57) = 0.25, *p* = 0.62, η^2^ = 0.00). There was a MCID (5.53) only in the nature condition. No score reached the level of high wellbeing on the SWEMWBS (27.5) at T2, although the nature condition had the highest value in absolute terms (26.67, see Table 1).

The NNT was 2.4 and 4.5 in the nature and indoor meditation groups, respectively, indicating that 2 to 5 young people would need to receive the intervention for one additional person to show 1 MCID in wellbeing.

### 3.3. Adherence to Meditation: A Dose–Response Analysis

There were positive associations between the number of times meditation was practiced and the increase in wellbeing for the nature condition (B = 0.77, SE = 0.30; β = 0.49, *p* = 0.02) but not the indoor group (B = 0.31; SE = 0.27; β = 0.22, *p* = 0.25). In the nature condition, every additional meditation attempt was associated with more than a half-point increase in the SWEMWBS. There was no significant association between the number of meditation attempts and depressive symptoms in the nature condition (B = −0.09; SE = 0.19; β = −0.08, *p* = 0.65) or the nature group (B = −0.56, SE = 0.30; β = −0.40, *p* = 0.08). Post hoc tests showed that the standardised residuals followed a normal distribution for the majority of models (W′ = 0.96 to 0.97, *p*s > 0.40), with the exception of the PHQ-8 model for the indoor condition (W′ = 0.89, *p* = 0.03). In addition, no Cook’s D scores were >1, nor were there any DFBETAs > ±1.

## 4. Discussion

This proof-of-principle study explored the potential for a brief 20 min self-guided audio recorded meditation to reduce depressive rumination and symptoms and improve mental wellbeing in a sample of university students, both immediately and over a two-week follow-up period. The results echo previous research in showing that a nature-based intervention can confer short-term mental wellbeing benefits [43,44,45]. While there was evidence of benefits following both meditations, the overall pattern in the data suggests that nature-based guided meditation may be superior to its non-nature-based indoor counterpart. More broadly, the plausibility of the positive effects of mindfulness-based approaches on young people, including brief interventions, is consistent with previous research [89,90,91], as is the finding that depressive rumination decreased following nature-based meditation when compared to active control and the indoor, standard meditation [46,65,66,67,68,70]. There are a number of possible explanations for the reduction in rumination. First, it may be that mindful meditation is augmented by the presence of a natural environment, which is consistent with research showing that an MBSR programme produced significant reductions in rumination in a natural outdoor setting [66]. Second, it may also be the case that the meditation simply provides a distraction from ruminative intrusive negative thoughts. While there is evidence to suggest that distraction can reduce rumination [92], this nevertheless seems a less plausible explanation given that research shows that mindfulness itself can reduce rumination [93] and that mindfulness reduces negative affects more than distraction alone [94].

A third possibility is that nature-based meditation activates or enhances processes that are mediated by the natural environment itself. For example, stress reduction theory [95] suggests that exposure to calming natural environments activates the parasympathetic nervous system and encourages positive changes in emotional states. Evidence from a meta-analysis has shown that exposure to natural environments does indeed reduce self-reported stress, salivary cortisol and blood pressure [39]. It is posited that with a reduction in stress comes a decrease in stress-reactive rumination [96,97], and so the fall in depressive rumination could be mediated by reductions in stress levels. This hypothesis was not testable in the current study but future studies should evaluate this possibility using experimental and longitudinal designs. Relatedly, it might also be the case that nature-induced reductions in rumination may be mediated by activation regions such as the subgenual prefrontal cortex (sgPFC) [68]. The sgPFC is an area that is activated when healthy individuals ruminate [98] and is implicated in major depression [99]. This is also critically related to cortisol secretion, the end product of the hypothalamic-pituitary adrenal (HPA) axis and predictive of de novo depressive episodes in adolescent boys [100]. Circumstantial evidence in animal models also points to a connection between the sgPFC and stress response. For example, in rhesus monkeys, glucose metabolism in the sgPFC has been shown to be strongly associated with plasma cortisol levels [101], which may indicate a causal role for the sgPFC in stress reduction and reduced rumination.

Fourth, attention restoration theory (ART) [102] suggests that limited cognitive attentional resources become depleted and are periodically in need of restoration [103]. Natural environments may help to replenish attentional resources through involuntary attention, a “soft fascination” with stimuli in the environment. Given that impaired attentional control can lead to increased rumination [104] it is plausible that attentional control is enhanced after exposure to nature-based meditation, which in turn limits perseverative negative thinking. ART also suggests that cognitive benefits from the natural environment may arrive via a number of other routes, including, but not limited to, self-reflection [105]. It is therefore likely that nature-based meditation facilitates positive self-reflection and positive rumination, thereby guarding against future depression. For example, in an ecological momentary assessment (EMA) study, the use of positive rumination was associated with more positive reported emotional experience [106].

Finally, Bratman and colleagues [70] suggest that nature contact may reduce rumination in two ways. First, aspects of the immediate environment themselves may disrupt the repetitive, perseverative negative thinking in the moment. Second, over longer periods of contact with nature, the tendency to ruminate may wane. Interestingly, our results for rumination in the nature-based condition suggest evidence for the former on the RRS. The latter mechanism may have been in operation in producing the beneficial effects at the two-week follow-up period on the depressive symptoms and wellbeing outcome measures. This conclusion is bolstered by the findings in the current study, the follow-up outcomes and adherence to meditation over the period.

While the student sample was not a clinical one, two measures of clinical relevance were calculated for wellbeing and depressive symptoms at follow-up; the MCID and the threshold cut off points for “high wellbeing” and “minimal depression”, respectively. For both depressive symptoms and wellbeing, 1 MCID was detected at T2 follow-up for the nature condition only and the group average score was close to but did not exceed the high wellbeing threshold (nature condition = 26.77; SWEMWBS high wellbeing = 27.50; see Figure 4). For depressive symptoms, both meditation groups reduced over time; in the nature condition there was a large effect size, whereas in the indoor group, there was a medium effect. The nature group average PHQ-8 score was below the threshold for depression (5 points) at follow-up, suggesting only minimal depression. The NNT in this study should be interpreted with caution but serves as an illustration of how many young people would need to complete the intervention for one of those to achieve a minimal clinically important improvement in depressive symptoms or wellbeing (NNT = 2–5). While these figures are often not comparable across studies, it is interesting to note that interventions designed to prevent cases of clinical depression in young people report NNTs in the order of 10 to 21 [107].

Practicing meditation as homework during the follow-up period was prospectively associated with positive outcomes; however, there was only evidence in support of this for the nature condition. The adherence analysis showed that for approximately every additional attempt at the nature-based meditation, wellbeing increased by more than half a point on the SWEMWBS. Although the association with depressive symptoms was not significant (*p* = 0.08) the same pattern was observed as for wellbeing but this was not true for the indoor standard meditation. More broadly the positive effect of practice on mental wellbeing is consistent with some previous research [108,109] whilst others have found no effect of practice [90]. It is likely that the benefits of meditation on mental wellbeing will build over the long-term, possibly in a non-linear way [110]. It is also important to recognise that the way in which young people are using meditation may moderate the effects on wellbeing. For example, a recent study showed that university students who used meditation in an attempt to control or avoid emotional experience reported more depressive symptoms than students whose intention to meditate was guided by acceptance [111].

### 4.1. Clinical Implications

Brief self-guided nature-based meditation may have different applications to clinical practice and has merit in both a prevention and intervention context [46]. Given findings from this study on reducing rumination and depressive symptoms and improving wellbeing, this form of intervention may have utility for individuals in beginning the therapeutic process whilst waiting to access treatment. Services could consider offering this after initial assessment as it lends itself to either individual or group delivery, depending on individual ability and potentially addressing service constraints by providing cost effective groups [112]. For example, although tested in a sample of university students, this intervention may have downwards and upwards extensions to both younger children, (who may benefit more from a group approach), and adults, who may be able to be self-led. Secondly, this intervention can also be used as an adjunct to most therapeutic approaches. This could enhance present-moment awareness, which facilitates insight and, therefore, is conducive to spotting maladaptive patterns, enabling behaviour and, subsequently, mood change [113]. Most therapeutic modalities contain an element of home practice to maintain the therapeutic process in between sessions (e.g., Cognitive Behavioural Therapy, MBIs); therefore, a brief nature-based meditation could be offered in this way to support awareness development [114]. Findings from this study demonstrate good acceptability by individuals to use this meditation independently in a home practice format. Finally, and in a similar way, this intervention could be written into individualised relapse prevention plans with the aim of maintaining a regular practice, or “dose” of nature-based meditation. This study showed incremental improvements in wellbeing for each session of meditation practice, which may mitigate relapse.

Recent adaptations to clinical practice during the COVID-19 lockdowns have necessitated in some cases that services move to outdoor spaces to continue therapeutic work. Providing the option for individuals to choose indoor or outdoor therapeutic spaces may aid engagement, rapport building and treatment adherence. A key question for further research will therefore be testing whether individual preference for NBIs has any impact on outcomes, compared with preferences for non-nature based interventions, as it is recognised that individuals may relate differently to the natural environment. This addresses the important question of “what works for whom?” [115] and aims to maximise therapeutic outcomes through personalised approaches. A further consideration in addition to individual preference for treatment space (whether in nature or indoors) is also whether potential added benefit may be gained for those individuals who are at risk of experiencing re-traumatisation [116]. Particular environments, such as typical clinical settings, can add to the experience of being retraumatised, due to perceived oppressive power imbalances and practices (e.g., locked doors, screens at reception). Clinicians could support individuals at risk of re-traumatisation by offering NBIs and sessions in nature to avert risks conferred through being in clinical buildings. Moreover, NBIs are increasingly being tested and developed in clinical psychology, with some holding the view that therapy outdoors may become part of mainstream training programmes and services [117].

### 4.2. Strengths, Limitations & Future Directions

The present study had a number of strengths, including using a randomised experimental design that included a comparison meditation condition as well as an active matched control condition, measures of adherence to the control condition and the meditation intervention over the two-week follow-up, the use of psychometrically validated and clinically relevant measures and a two-week follow-up period to assess the short-term effects of the meditations. A good retention rate was also achieved (88%) at follow-up. However, the study was underpowered to detect smaller effects, which may explain the lack of statistical significance in the BSRI analysis. Future studies should aim to increase the sample size. This would also allow for further hypotheses to be tested. For example, personalised approaches may enhance efficacy. It might be that the intervention is particularly relevant for individuals with different levels of nature connectedness or mindfulness. Although we found that the nature-based, outdoor meditation was generally more effective in this study, it is not entirely clear whether this was due to the mindful attention paid to nature or to therapeutic mental imagery being generated [118]. We therefore recommend that future studies compare the effect of *in natura*, nature-based meditation with guided nature-based imagery alone. Future work should also test whether there is an additional benefit to nature-based meditation compared to simple exposure to natural environments. As pointed out earlier, approaches such as forest bathing often have a mindful component embedded within the programme and so there is an issue here of confounding. Nevertheless, appeal to experimental methods will allow the creation of appropriate conditions to test the hypothesis (e.g., nature-walking whilst meditating vs. problem-solving). Finally, we do not underestimate the complex nature of mental wellbeing and recognise that other factors, including distal ones such as early adversity and proximal events such as exam stress, may influence individual differences in rumination. Similarly, access to green or blue space or meditation background may influence these outcomes. Although differences between groups on these variables are unlikely with random allocation to condition, such factors may moderate the effect of these interventions. Future work should test these and other hypotheses with larger sample sizes.

## 5. Conclusions

While the evidence in the present study suggests that there are benefits to this very brief meditation intervention, we should be cautious and not overly ambitious on claims of brief MBIs [119]. This study provided preliminary evidence for the superiority of nature-based meditation relative to the indoor condition. This intervention may have scope for a broad application, such as utility before therapy whilst on a waiting list, as an adjunct of therapy, and as part of a relapse prevention plan. The study also found support for practicing meditation in the nature condition, which prospectively predicted positive outcomes. The findings, particularly on novel nature-based meditation, should be replicated in a larger sample and tested in clinical samples before firmer conclusions are made on its efficacy.

## Figures and Tables

**Figure 1 ijerph-19-09118-f001:**
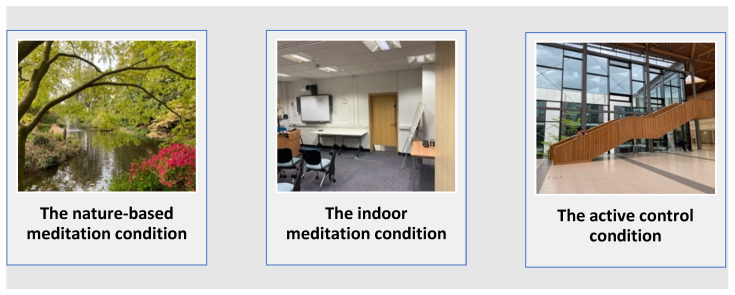
The experimental condition locations.

**Figure 2 ijerph-19-09118-f002:**
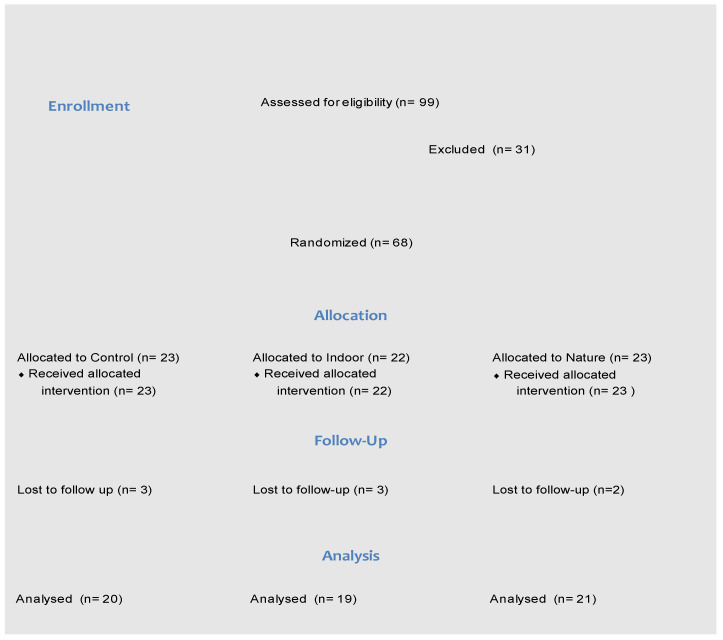
CONSORT flow chart.

**Figure 3 ijerph-19-09118-f003:**
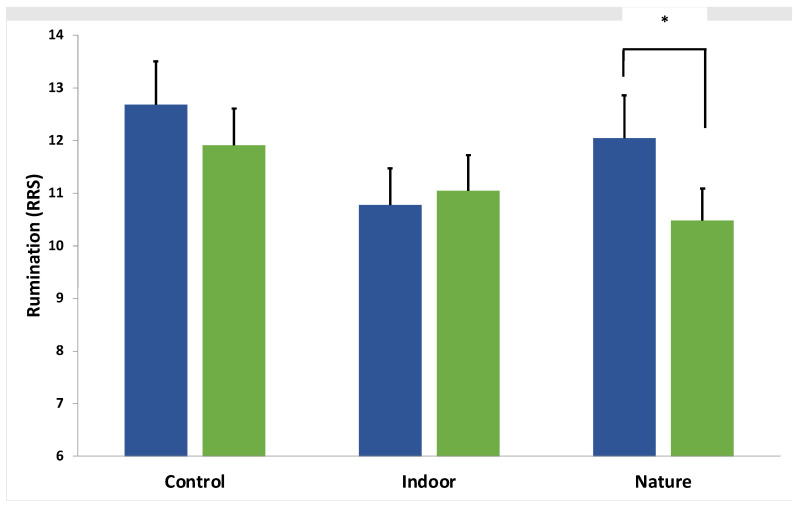
Group × Time interaction for trait rumination (RRS). Note. * indicates a significant decrease in rumination (*p* < 0.05). Blue and green bars represent the group average at T0 and T1, respectively.

**Figure 4 ijerph-19-09118-f004:**
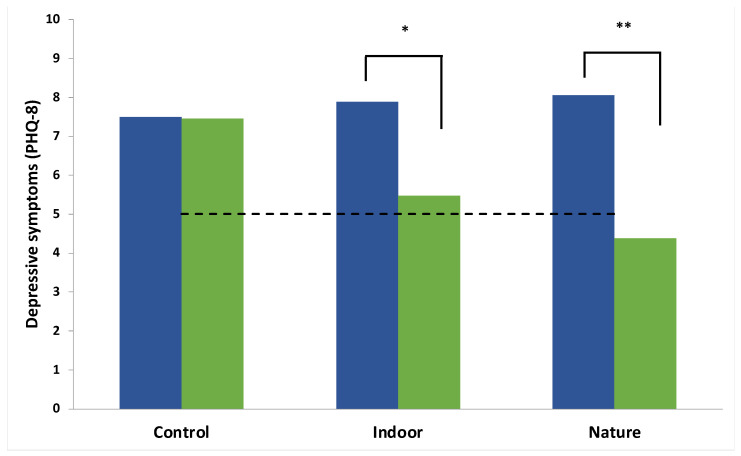
Group × Time interaction for depressive symptoms (PHQ-8). Blue and green bars represent the group average at T0 and T2, respectively. Note. * indicates a significant decrease in depressive symptoms (*p* < 0.05). ** indicates a significant decrease in depressive symptoms (*p* < 0.01). The dashed horizontal line refers to the cut point for minimal depression.

**Figure 5 ijerph-19-09118-f005:**
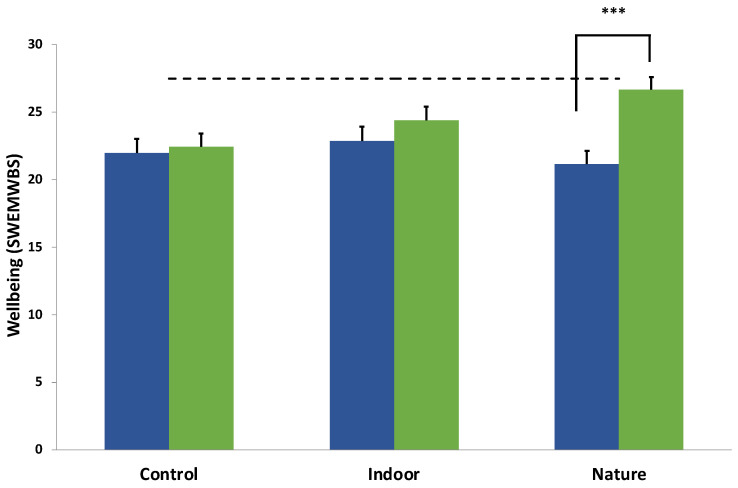
Group × Time interaction for Wellbeing (SWEMWBS). Note. *** indicates a significant decrease in depressive symptoms (*p* < 0.001). The dashed horizontal line refers to the cut point for high wellbeing. Blue and green bars represent the group average at T0 and T1, respectively.

**Table 1 ijerph-19-09118-t001:** Alignment to Gaussian distribution before and after square root transformation using the Shapiro-Francia test (W′).

Variable	Untransformed Distribution	Square Root Transformation
RRS T0	W′ = 0.97, *p* = 0.14	-
RRS T1	W′ = 0.97, *p* = 0.12	-
BSRI T0	W′ = 0.98, *p* = 0.48	-
BSRI T1	W′ = 0.99, *p* = 0.58	-
PHQ-8 T0	W′ = 0.93, *p* < 0.001	W′ = 0.99, *p* = 0.84
PHQ-8 T2	W′ = 0.93, *p* < 0.001	W′ = 0.99, *p* = 0.68
SWEMWBS T0	W′ = 0.99, *p* = 0.93	-
SWEMWBS T2	W′ = 0.99, *p* = 0.66	-

**Table 2 ijerph-19-09118-t002:** Descriptive statistics at baseline post intervention and follow up and simple main effects of group.

Outcome	T0	Baseline Differences (*p*-Value *)	T1	T2	Simple Main Effects of Group **
	Mean (sd)		Mean (sd)	Mean (sd)	
RRS		0.25			
Nature	12.04 (3.90)		12.04 (3.90)	10.48 (2.94)	**F(1,65) = 11.14, *p* = 0.001, η^2^ = 0.15**
Indoor	10.77 (3.29)		10.77 (3.29)	11.05 (3.20)	F(1,65) = 0.32, *p* = 0.57, η^2^ = 0.01
Control	12.57 (3.90)		12.57 (3.90)	11.87 (3.20)	F(1,65) = 2.2, *p* = 0.14, η^2^ = 0.03
BSRI		0.64			
Nature	357.65 (180.38)		269.04 (146.61)		**F(1,65) = 11.79, *p* = 0.001, η^2^ = 0.15**
Indoor	325.45 (166.96)		307.27 (155.35)		F(1,65) = 0.48, *p* = 0.49, η^2^ = 0.01
Control	374.78 (183.08)		356.00 (205.49)		F(1,65) = 0.53, *p* = 0.47, η^2^ = 0.01
PHQ-8		0.85			
Nature	8.05 (4.66)			4.38 (4.48)	**F(1,57) = 13.94, *p* < 0.001, η^2^ = 0.20**
Indoor	7.89 (5.23)			5.47 (2.55)	**F(1,57) = 5.50, *p* = 0.02, η^2^ = 0.09**
Control	7.50 (5.30)			7.45 (5.54)	F(1,57) = 0.00, *p* = 0.96, η^2^ = 0.00
SWEMWBS		0.38			
Nature	21.14 (4.16)			26.67 (5.03)	**F(1,57) = 40.10, *p* < 0.001, η^2^ = 0.41**
Indoor	22.89 (4.38)			24.42 (3.52)	F(1,57) = 2.78, *p* = 0.10, η^2^ = 0.05
Control	22.00 (5.17)			22.45 (4.14)	F(1,57) = 0.25, *p* = 0.62, η^2^ = 0.00

* ANOVA. ** Simple main effects compare the change over time within a condition. The purpose is to decompose and illustrate the pattern in the data following a significant Group × Time interaction. Bold entries are statistically significant changes (*p* < 0.05). For comparison and completeness, the non-significant finding for BSRI is also included in the table. RRS = Ruminative Response Styles; BSRI = Brief Rumination State Inventory; PHQ-8 = Patient Health Questionnaire; Short Warwickshire Edinburgh Mental Wellbeing Scale.

## Data Availability

Please contact the authors directly to request datasets for the study.

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
