# Peer review of "Nature-Based Meditation, Rumination and Mental Wellbeing"

_ijerph, 2022, doi:10.3390/ijerph19159118_

Round 1
Reviewer 1 Report
Thank you for the possibility to review this interesting manuscript.
The manuscript “Nature-based meditation, rumination and mental wellbeing” presents results on effects of minimal nature-based and indoor meditation interventions on mental health in a sample of young adults. Information on minimal interventions for the prevention of mental health problems are significant for clinical practice. Looking at nature-based interventions are highly relevant and may be specifically well accepted by young people.
The aim of the study and the reported results are well stated. The manuscript is generally well-written and focuses on an interesting and important topic.
I have a few comments for the authors to consider in order to improve the manuscript further:
· There are a few spelling mistakes for the word meditation. Please check again the abstract (line 12) and the introduction (p. 3, line 153)
Introduction
· First time an abbreviation is used in the text the term should be written out (e.g. see line 65 – abbreviation cbt/ ipt) as potential readers may come from different educational or research backgrounds.
· The introduction is well-written and very comprehensive. The information presented here is important and connected to the objective of the study; however, in my opinion the introduction and overall paper would benefit from a condensation of the information given in the introduction; in particular the following sections: lines 62 – 75, lines 90 – 108, lines 113 – 122, 123-136 may benefit from a further condensation.
Materials & Methods
· I was wondering if there is a study protocol and whether the study was registered beforehand, as it is a randomized experimental design? I could not find information on that in the manuscript.
· I would suggest that the authors include the abbreviations/terms T0, T1 & T2 in the methods section 2.8 at the respective position for better clarity of the three assessment points.
· Did participants get any incentive for their participation?
· Have inclusion and/or exclusion criteria been specified before recruitment?
· Sample size: Did the sample size calculation of N=105 include the consideration of dropouts? I was further wondering, what the reasons were that only 99 instead of 105 participants have been recruited in the first place. Were there difficulties in recruiting enough students for the study?
· The section 2.6 Measures should include information on what further aspects apart from the primary outcomes have been assessed, eg. sociodemographics/ living situation etc.
· I was wondering if information on the following aspects have been assessed as well:
o Differences in university demands between students during study participation
o Access and usage of green space during the two weeks of study participation independent of group allocation
· When using abbreviations > first time should be written out (MCID, p.6, line 274)
· How often was the control group encouraged to walk around the Forum during study participation?
Results
· Figure 1: There are no lines or arrows leading through the flow chart > is it supposed to be like that?
· Tables should be self-explanatory. I would suggest including a legend for the abbreviations used in the tables.
· Similarly Figure 3/4 & 5: different colors for two time points (blue & green) should be explained in each figure for better clarity.
· Section 3.3 Adherence to meditation: line 385 - What is the outdoor group if it is not the nature group?
Discussion
The Discussion is well-written and the results are discussed in detail under consideration of the existing literature. In the discussion it is mentioned that exposure with nature itself may reduce rumination. With this in mind, another active control group with nature exposure only (without the meditation) may have been interesting here. Maybe meditation is not needed at all to improve mental health outcomes if time is spend in nature.
Limitations
It should be mentioned that other factors influence depression, rumination and wellbeing, e.g. anxiety, upcoming exams etc., which have not been controlled for in the study.
Similarly, if usage and access to greenspace has not been assessed it should be mentioned in the limitations as it may also act as a confounder.
Reviewer 2 Report
Authors’ English is of high quality. The paper requires only some editorial amendments but not many linguistic corrections. For example, in same cases the word ‘mediation’ instead of ‘meditation’ is used (lines: 12, 153, 473, 477, 479 and 535).
The literature is relevant and up-to-date as well as the sources are rich; however, authors should carefully verify the correctness of the notation of the sources, because some of them require amendments, such as, for example, item 42 (i.e. Koteral et al.), where instead of the range of pages 1-25, there should be 337–361, or the item 53 (i.e. Umegaki et al.), where there is no numbering of the quarterly issue and the page range (it should be 29(2), 468–484; see the website: https://www.sciencedirect.com/science/article/pii/S1077722921000225).
The four research hypotheses sound correct, but the composition of the ‘Results’ section seems not fully clear and adequate when we mean its logical reference to the previously formulated research hypotheses. The results are presented generally in a clear way. However, Figure 1 is visually somewhat chaotic/atomized. Perhaps adding some kind of arrows would solve the problem. The conclusions are too short and superficial, and therefore they need elaboration.
The comma before ’42 females’ is missing (line 156).
The word ”also” was used twice in the sentence: “ART also suggests that cognitive benefit from the natural environment may also arrive via a number of other routes” (lines 436–437).
